# Antiviral Activity of Anthranilamide Peptidomimetics against Herpes Simplex Virus 1 and a Coronavirus

**DOI:** 10.3390/antibiotics12091436

**Published:** 2023-09-12

**Authors:** Umme Laila Urmi, Samuel Attard, Ajay Kumar Vijay, Mark D. P. Willcox, Naresh Kumar, Salequl Islam, Rajesh Kuppusamy

**Affiliations:** 1School of Optometry and Vision Science, University of New South Wales, Sydney, NSW 2052, Australia; v.ajaykumar@unsw.edu.au (A.K.V.); salequl@juniv.edu (S.I.); r.kuppusamy@unsw.edu.au (R.K.); 2School of Chemistry, University of New South Wales, Sydney, NSW 2052, Australia; s.attard@unsw.edu.au (S.A.); n.kumar@unsw.edu.au (N.K.); 3Department of Microbiology, Jahangirnagar University, Savar 1342, Bangladesh

**Keywords:** antiviral compounds, peptidomimetics, MHV-1, HSV-1, envelope disruption

## Abstract

The development of potent antiviral agents is of utmost importance to combat the global burden of viral infections. Traditional antiviral drug development involves targeting specific viral proteins, which may lead to the emergence of resistant strains. To explore alternative strategies, we investigated the antiviral potential of antimicrobial peptidomimetic compounds. In this study, we evaluated the antiviral potential of 17 short anthranilamide-based peptidomimetic compounds against two viruses: Murine hepatitis virus 1 (MHV-1) which is a surrogate of human coronaviruses and herpes simplex virus 1 (HSV-1). The half-maximal inhibitory concentration (IC_50_) values of these compounds were determined in vitro to assess their potency as antiviral agents. Compounds **11** and **14** displayed the most potent inhibitory effects with IC_50_ values of 2.38 μM, and 6.3 μM against MHV-1 while compounds **9** and **14** showed IC_50_ values of 14.8 μM and 13 μM against HSV-1. Multiple antiviral assessments and microscopic images obtained through transmission electron microscopy (TEM) collectively demonstrated that these compounds exert a direct influence on the viral envelope. Based on this outcome, it can be concluded that peptidomimetic compounds could offer a new approach for the development of potent antiviral agents.

## 1. Introduction

The outbreak of the COVID-19 pandemic in 2019/2020 focused the world’s attention on viral infections and the ease with which they can rapidly spread around the world. According to World Health Organization data, there have been at least 769,774,646 cumulative cases and 6,955,141 cumulative deaths from the SARS-CoV-2 virus that is the cause of COVID-19 (https://covid19.who.int, accessed on 19 August 2023) as of 16 August 2023. COVID-19 remains endemic in many countries in the world [1]. The coronaviruses are not the only cause of viral pandemics; the influenza virus was also found to cause pandemics which have been as large and severe as COVID-19 [2]. Other viruses like Herpesviridae can cause a variety of diseases including cold sores [3], genital herpes [3], and varicella-zoster (also known as chickenpox) [4]. Noroviruses are leading causes of vomiting, diarrhea, and foodborne illness [5]. However, there is a paucity of antiviral drugs that can protect against or treat many of these diseases. Hence, there is a need to develop new antiviral drugs.

The beta coronaviruses [6], other than SARS-CoV-2, SARS-CoV, and MERS-CoV, have been linked to severe acute respiratory syndrome in humans. A mouse beta coronavirus called murine hepatitis virus (MHV) is often used as a surrogate for the human coronaviruses [7], being recommended as a surrogate by Therapeutic Goods Australia (TGA) (https://www.tga.gov.au/resources/resource/guidance/surrogate-viruses-use-disinfectant-efficacy-tests-justify-claims-against-covid-19, accessed on 19 August 2023). Phylogenetic analysis has revealed that SARS-CoV and SARS-CoV-2 are part of the betacoronavirus lineage b, MERS-CoV belongs to lineage c, and MHV falls under lineage a [6]. This places MHV as a virus more closely related to the SARS-associated coronaviruses than other potential surrogates such as human coronavirus 229E, feline coronavirus or the transmissible gastroenteritis virus, which are alphacoronaviruses belonging to different lineages [6]. Unlike human SARS-CoV, SARS-CoV-2 and MERS-CoV that require biosafety level 3 containment, MHV requires a biosafety level 2 laboratory for propagation and safe handling [8]. MHV has been used as a model to investigate the effects of temperature and humidity on virus survival [9] and the efficacy of common disinfectants [10,11]. MHV has also been employed in evaluating the effectiveness of certain antiviral candidates [12,13,14,15,16,17,18,19]. Currently, treatments for SARS-CoV-2 infection include molnupiravir, nirmatrelvir, ritonavir and remdesivir, but these are only recommended for people at high risk of severe infection [20].

HSV-1 is an enveloped virus containing a double-stranded DNA [21]. HSV-1 infection remains endemic in more than half of the global population and reactivates periodically giving rise to a worldwide health issue [22]. In developed nations, HSV-1 infection leads to recurring oral sores, and serves as the primary reason behind cases of infectious blindness and genital infections [3]. Most HSV-1 infections typically happen in childhood and once the person has been infected, the virus remains in the body indefinitely, with the possibility of that person’s experiencing viral shedding, whether with symptoms or without, throughout their life [21]. Acyclovir and its derivates can be used to treat HSV-1 infections [23]. However, the emergence of drug-resistant strains within the HSV-1 population, particularly those exhibiting resistance to acyclovir [24,25], has introduced substantial impediments to successful therapeutic interventions. In this era dominated by discussions of COVID-19 and its associated coronaviruses, it is imperative to recognize the importance of other viral families such as Herpesviridae, represented by HSV-1. In recent years, the exploration of alternative antiviral agents has gained momentum, with a particular focus on peptides and peptide mimics as promising candidates [26].

Antimicrobial peptides (AMPs) have attracted much attention due to their antiviral properties together with their antibacterial activity [27]. Many AMPs have inhibitory activities against coronaviruses, HSV, influenza virus, dengue virus, zika virus, hepatitis C virus, etc. [27]. The antiviral activities of AMPs and their mechanism of action vary depending on their structural characteristics such as the α-helix, β-sheet, cyclic β-sheet, β-turn and extended peptide [27,28]. A recent comprehensive review delved into the progress made in creating antiviral drugs using peptides and related compounds [26]. Despite the substantial promise of the antimicrobial peptides, they come with constraints, such as inadequate availability within the body, the potential to trigger an immune response, toxicity, stability concerns, and a high cost of production [29].

The justification for developing peptidomimetics, which are mimics of antiviral peptides, stems from the drawbacks associated with AMPs. Peptide mimetics are synthetically generated compounds designed to replicate both the structural and functional characteristics of naturally occurring peptides [30]. These fabricated molecules are engineered to mirror fundamental aspects of peptides. They can be formulated to achieve optimal bioavailability, ensuring that they have effective concentration at specific sites within the body [31]. Customizing peptide mimetics can give them heightened selectivity for viral targets, thereby minimizing any potential impact on healthy host cells [32]. This specificity enhances therapeutic efficacy while reducing the likelihood of adverse effects. Additionally, mimetics can be tailored to interact with various types of viruses, offering a broader range of antiviral activity [32]. In comparison to the potential difficulties and costs linked to the large-scale production of natural peptides, the synthesis of peptide mimetics can often be achieved using more efficient and cost-effective techniques. This streamlined production process contributes to their viability for mass production [33]. The approaches for transforming peptides into peptidomimetics within the realm of antiviral applications involve tactics such as altering terminal structures [34], substituting amide bonds with isosteric counterparts at particular locations [35], modifying amino acids [36], implementing an inverse-peptide strategy [37], and employing a cyclization strategy [38].

A variety of synthetic compounds containing cationic and hydrophobic side groups have structural and functional similarities to AMPs. Recent preliminary investigations have indicated that peptoids MXB004, MXB005, and MXB009, sequence-specific *N*-substituted glycine oligomers, exhibit potent in vitro antiviral activity against HSV-1 and SARS-CoV-2 [39]. Cryo-electron microscopy (Cryo-EM) images have revealed substantial disruption of viral envelopes. However, further research is necessary to understand the interactions between these mimetics and the components of the viral envelope [39]. In addition, the previous study only examined whether the peptoids interacted directly with the viral particles, whereas it is known that AMPs can act not only on viral particles but also by preventing viral attachment to host cells or disrupting viral replication within host cells [27].

Anthranilamide possesses a special chemical configuration characterized by amide functional groups which makes it a useful building block for the synthesis of organic compounds [40]. Prior investigations have documented that derivatives derived from anthranilamides have inhibitory effects against bacteria [40], can disrupt bacterial biofilms [40], and also have antiviral properties against alphaviruses [41]. The objective of this study was to investigate the antiviral effectiveness of anthranilamide-based peptide mimetics against two distinct viral pathogens, MHV-1 and HSV-1. These viruses have distinct genomes, diverse viral entry pathways, and varied pathologies within a host. The antiviral activity of 17 mimetics against enveloped viruses was investigated. Additionally, the envelope-mediated antiviral activity of these mimetics was explored using transmission electron microscopy.

## 2. Results and Discussion

### 2.1. Screening of Anthranilamide Peptidomimetics against MHV-1 and HSV-1

Three different assays were performed to understand the mode of action of the peptidomimetic compounds on MHV-1 and HSV-1. The three different assays used were virus pre-treatment, cell pre-treatment, and treatment of cells post-viral infection [42], with the main distinction being the timing of the compound addition [43]. These assays help us to understand whether the compounds can act directly on viral particles, interact with the target cell and prevent the subsequent binding to the virus surface, or interfere with the viral replication.

At first, all the compounds were tested by direct virus inactivation method. IC_50_ values obtained for the **1**–**17** compounds against MHV-1 and HSV-1 are summarized in Table 1. The tested compounds **1**–**5** with different halogen and methoxy substituents and containing an amino cationic group demonstrated no activity even at 62.5 μM against MHV-1 and HSV-1. Substituting the amino groups or guanidines produced compounds **6**–**10**. None of these compounds was active against MHV-1. However, the compound that included a bromo substituent, **7**, was an active antiviral against HSV-1 at 32.7 µM. Substituting the bromo with a chloro substitution improved the antiviral activity to 14.8 µM against the same virus.

Subsequently, the effect of changing hydrophobic groups was examined by moving these groups from second position to fifth position of the anthranilamide backbone. Initially, the same naphthyl hydrophobic group used in **1**–**10** was used to replace the bromo group to yield compound **11**. Excellent activity was observed for compound **11** against MHV-1 (IC_50_ 2.38 µM) and good activity for HSV-1 (IC_50_ 34.9 µM). The substitution of these naphthyl groups with biphenyl groups yielded the corresponding isosteres, compounds **12**–**14**. Of these, compound **14** showed improved activity against HSV-1 (IC_50_ 13 µM) while maintaining good activity against MHV-1 (IC_50_ 6.3 µM). Synthesis of the compounds with biphenyl or naphthyl hydrophobic groups with lysine as the cationic group yielded compounds **15**–**17**. While these showed activity against MHV-1 of IC_50_ **11** to 22 µM, they were not active against HSV-1.

Therefore, compounds **11** and **14** showed good activity against both viruses and this can be seen in Figure 1 and Figure 2 outlining the detailed antiviral impacts of these two compounds at varying concentrations against MHV-1 and HSV-1, respectively.

Comparison of all the IC_50_ values of the compound showed that the placement of hydrophobic group in the anthranilamide backbone plays a key role in the antiviral activity against different viral families, MHV-1 and HSV-1. The results contrast with compound **12** with the 4-biphenyl substitution which showed no activity even at 62.5 µM. Perhaps, apart from hydrophobicity, the orientation of the compound may play a role in antiviral activity. This discrepancy in antiviral activity across different viral families underscores the importance of investigating the specific mechanisms of action for these compounds.

In order to explore additional potential mechanisms, the ability of compounds **11** and **14** to reduce the ability of the two viruses to bind to their respective mammalian host cells was examined. Three concentrations (15.625 µM, 7.5 µM, and 3.75 µM) of compounds **11** and **14**, all below their cytotoxic thresholds, were allowed to interact with A9 and Vero cells for 24-h before virus infection. The impact of these interactions was assessed using a plaque assay to determine inhibitory effects. Notably, none of the compounds could inhibit virus infection when they were preincubated with the cells (Figure 3A–D).

In order to determine if compounds **11** and **14** were able to inhibit aspects of viral infection after the initial binding stage to host mammalian cells, the same concentrations of the compounds were administered to the cells after a 3-h interval post-infection. In this assay, no noticeable antiviral activity was observed (Figure 3A–D). This investigation strongly suggests that the compounds were primarily targeting the viruses themselves.

### 2.2. TEM Images

The results from the previous experiments suggested that the mimics directly target the viruses. Therefore, the compound with the highest activity (lowest IC_50_) was incubated with the appropriate virus and the effect examined by transmission electron microscopy. Compound **11** treated MHV-1 (Figure 4) showed a disrupted viral envelope compared to the control samples containing untreated viruses in PBS. In the control images, the viral envelope and spike proteins were clearly visible. However, in the treated samples, the lipid envelope was completely disrupted, leaving the naked capsid exposed. For compound **14**, with HSV-1, there was a partial disruption of the HSV-1 envelope in treated samples, while the control samples remained intact (Figure 5).

These findings demonstrate that both compounds disrupted viral envelopes, and this was the most likely reason for the reduction in viral infectivity. This is like the reported effects of peptoids and several AMPs [27,39]. MHV-1 most likely obtains its viral envelope from the Golgi apparatus [44] much like other coronaviruses [45,46]. On the other hand, HSV-1 obtains its envelope by more complex mechanisms that involve the inner and outer nuclear membranes, membranes of organelles and the plasma membrane of host cells [47]. While direct comparisons of the lipids of the envelopes of MHV-1 and HSV-1 are lacking, it is likely that they contain different lipids due to differences in their assembly, and this may be the reason for differences in interactions with the various peptidomimetics in the current study.

While enveloped viruses acquire lipids from host cells, the composition of viral membranes can significantly differ from that of the host [48]. The lipid diversity within viral envelopes is an emerging area of study, with ongoing lipidomic investigations shedding light on the importance of diverse lipids in viral infections [49]. The viral envelope’s lipid composition includes various lipids such as phosphatidylcholine (PC), phosphatidic acid, phosphatidylglycerol, sphingolipids, phosphatidylethanolamine, and phosphatidylserine (PS)—each present in concentrations distinct from host cells [48]. Enveloped viruses often exhibit elevated PS levels on their outer surface compared to host cell membranes, facilitating engagement with PS-mediated pathways for cellular entry [50]. These disparities make envelope-based PS an attractive and specific target when designing treatments for enveloped viruses [51]. These differences may also allow the production of peptidomimetics that can specifically target the viral envelope while being inactive, or less active, against the plasma membrane lipids of host cells.

### 2.3. Cytotoxic Activities of Active Mimetics

The cytotoxic potential of the peptidomimetics was assessed using A9 and Vero cells. The concentrations of peptidomimetics required to inhibit 50% of virus infectivity (IC_50_) and induce 50% cell toxicity (CC_50_) over a specific timeframe were measured. The therapeutic index (TI), calculated as the ratio of CC_50_ to IC_50_, was determined. The outcomes for all analogs are presented in Table 2 against both virus strains.

Earlier research with AMPs had indicated that increasing the overall positive charge and moderating hydrophobic characteristics resulted in decreased cytotoxic activity and heightened efficacy against bacteria [52]. In the context of A9 cells, the compounds showing the least cytotoxicity were compounds **16** and **17** with biphenyl groups (Table 2). This was followed by compound **15** that contained a lysine cationic group. As compound **17** also had a low IC_50_, this resulted in a good therapeutic index of 9.71, i.e., the concentration that killed 50% of the A9 cells was at least nine times higher than the concentration needed to prevent 50% of MHV-1 cells from infecting the A9 cells. In the case of Vero cells, compound **7** containing a guanidine cationic group with a halogen substitution gave low cytotoxicity and the highest therapeutic index of 3.87. However, this compound was relatively less active against the viral cells (Table 2). Across both cell types, compounds **11** and **14**, which contained amino substitutions and the hydrophobic groups naphthyl and biphenyl, respectively, displayed a similar cytotoxicity to the A9 and Vero cells.

## 3. Materials and Methods

### 3.1. Chemistry

The general route for the preparation of anthranilamide peptidomimetics **1**–**17** is presented in Figure 1 and Figure 2.

The synthesis presented in Figure 1 and Figure 2 gives a robust and efficient synthetic sequence to access a series of guanidine-containing peptidomimetics [38,39]. The strategy involves a ring-opening reaction of various substituted isatoic anhydrides with tert-butyl (S)-(2-(2-amino-3-(1H-indol-3-yl)propanamido)ethyl)carbamate, yielding an intermediate compound. Subsequent reaction with naphthoyl chloride and deprotection of the tert-butyl (BOC) group produced amine compounds (**1**–**5**) and guanidine-containing compounds **6**–**10** on treatment of 1–5 with N,N′-Bis-Boc-1-guanylpyrazole. Further diversification of the intermediates was accomplished through Suzuki coupling, involving intermediate compound **1A** with diverse boronic acids, followed by BOC group deprotection, leading to the successful synthesis of compounds **11**–**14**. Additionally, compound **1A** underwent transformation via reaction with various acid chlorides, subsequent deprotection, and treatment with Nα,Nε-Di-Boc-L-lysine hydroxysuccinimide ester, thereby affording the formation of a series of amine compounds **15**–**17** upon BOC group deprotection. The detailed procedure of the intermediate and the final compounds are presented in references [40,53]. The Appendix A contains the data for all the final compounds. All the compounds from **1**–**17** were tested for activity against HSV-1 and MHV-1.

All chemical reagents were purchased from commercial sources (Combi-Blocks (San Diego, CA, USA), Chem-Impex (Wood Dale, IL, USA), Thermo Fisher Scientific (Waltham, MA, USA), and Sigma Aldrich (St. Louis, MO, USA)) and used without further purification. The solvents were commercial and used as obtained. The reactions were performed using oven-dried glassware under an atmosphere of nitrogen and in anhydrous conditions (as required). Room temperature refers to the ambient temperature. Flash chromatography and silica pipette plugs were performed under positive air pressure using Silica Gel 60 of 230–400 mesh (40–63 μm) and using Grace Davison LC60A 6-μm for chromatography. Proton and Carbon NMR spectra were recorded in the solvents that were specified using a Bruker DPX 300 or a Bruker Avance 400 or 600 MHz spectrometer as designated. ^1^H NMR spectroscopic data are reported as follows (chemical shift in ppm; multiplicity in br, broad; s, singlet; d, doublet; t, triplet; q, quartet; quint, quintet; sext, sextet; sept, septet; m, multiplet; or as a combination of these (e.g., dd, dt, etc.)); coupling constant (J) in hertz, integration, proton count, and assignment.

### 3.2. Analytical Data

All the final compounds ^1^H and ^13^C NMR data are presented in the Appendix A.

### 3.3. Virus and Cell Culture

An American type culture collection (ATCC) strain of the mouse hepatitis virus (MHV) ATCC/VR261 and herpes simplex virus type 1 (HSV-1) was grown in A9 ATCC/CCL 1.4 [11] and Vero cells [54], respectively, in Dulbecco’s minimum essential medium (DMEM) (Thermo Fisher Scientific (Waltham, MA, USA) containing 10% fetal bovine serum (FBS) and 1% antibiotics (streptomycin sulphate and penicillin G). Virus stock was prepared and stored at −80 °C.

### 3.4. Antiviral Testing

The peptidomimetics were evaluated for their antiviral activity using three different assays: virus pre-treatment [42], cell pre-treatment [42], and post-treatment assays [42], with the main distinction being the timing of the compound addition [43]. Cells were seeded in 24-well plates at a density of 2.5 × 10^5^ cells/well and incubated for 48 h at 37 °C. In all assays, the compounds were added to DMEM medium in the absence of fetal bovine serum (FBS). All experiments were performed in duplicate with two independent replicates. The inhibitory effect on viral infectivity was determined by comparing the number of plaques in wells treated with the test compounds (peptidomimetics) to the plaques observed in the control wells (cells infected with the virus without test compounds).

Virus pre-treatment assay: In this assay, the test compounds were added to the virus (1 × 10^3^ PFU/mL) and incubated for 3 h at 37 °C. After incubation, each mixture (virus/peptide) was diluted in DMEM medium containing 10% FBS to neutralize the peptides or mimics [55]. The dilutions were then added to the cell monolayers for 1 h, followed by the addition of an agar overlay containing 1% agar. The cells were further incubated for 72 h, after which they were fixed, stained, and the number of plaques was counted.

Cell pre-treatment assay: The cells were treated with the test compounds and incubated for 24 h at 37 °C. Subsequently, the virus was added and allowed to infect the cells for 1 h at 37 °C. After that, the excess virus was discarded and an agar overlay was added, and the cells were incubated for another 72 h at 37 °C. The cells were then fixed, stained, and the number of plaques was determined.

Post-viral infection assay: The cells were first incubated with viruses (1 × 10^3^ PFU/mL) for 3 h at 37 °C to ensure complete cellular entry. Following that, the test compounds were added and incubated for 24 h at 37 °C. Finally, compound solutions were discarded from wells and an agar overlay was added, and the cells were incubated for another 72 h at 37 °C. The cells were then fixed, stained, and the number of plaques was determined.

The assay that exhibited the highest antiviral activity for most of the viral models tested was selected, and a non-linear regression analysis was performed using GraphPad Prism software to determine the IC_50_ (concentration required for 50% inhibition of a specific target).

### 3.5. Cytotoxicity Assay

The MTT assay [43,52,56] was used to assess the cytotoxicity of compounds against the viral host (A9 and Vero) cells. This assay relies on the reduction of the yellow MTT compound to a dark blue formazan product by viable and metabolically active cells. Viral host cells were cultured in 96-well plates and incubated at 37 °C in a 5% CO_2_ atmosphere, treated with various concentrations of the test compounds together with further incubation for 24 h. After incubation, 100 μL of a 5 mg/mL MTT solution was added to each well and incubated for 2–4 h at 37 °C. The supernatant was then discarded, and 100 μL of 100% DMSO was added to dissolve the formazan salts with vigorous agitation for 10 min at room temperature. The absorbance at 540 nm was measured using a spectrophotometer. Cytotoxicity was determined by comparing the absorbance values in the test and control wells expressed as a percentage. A total of 100 μL DMSO was used for negative control (ctr−), while 100 μL of culture medium represented the positive control (ctr+). All experiments were repeated three times, and the mean values with standard deviations are reported. Nonlinear regression analysis was performed using GraphPad Prism software to determine the CC_50_ (concentration at which 50% of cells are killed).

### 3.6. Transmission Electron Microscopy

The viruses were treated with compound **11** for MHV-1 and **14** for HSV-1, both at 37 °C for 2 h. Subsequently, 10 µL of the treated mixture was placed on a glow-discharged carbon-coated 200 mesh copper grid and allowed to evaporate for 5 min. The sample was then stained with 1% phosphotungstic acid (pH 6.5) for 30 s to improve the contrast. After air-drying, the grids were examined using an FEI Tecnai G2 20 TEM machine [57,58,59].

### 3.7. Statistical Analysis

The data are presented as the mean ± standard deviation (SD). Data analysis was performed using a non-parametric Kruskal–Wallis test with Dunn’s multiple comparisons. A significance level of *p* < 0.05 was considered statistically significant.

## 4. Conclusions

In summary, novel short peptidomimetics derived from anthranilamide compounds were synthesized by altering their hydrophobic group’s properties and position, together with the cationic charge of the molecules. This led to the creation of compounds displaying good antiviral effects against MHV-1 and moderate effects against HSV-1. The toxicity levels of these compounds varied when tested on cells; compounds **11** and **17** had the highest therapeutic index for MHV-1, while compounds **7** and **9** had greater TI values for HSV-1. Compound **11** gave over 50% inhibition of MHV-1 at 2.18 µM, and compound **14** had over 50% inhibition of HSV-1 at 13 µM. Analysis using transmission electron microscopy demonstrated that the active compounds could induce the disruption of viral envelopes, resembling the behavior of AMPs. Consequently, this category of peptidomimetic compounds introduces a fresh avenue for the development of potent antiviral agents.

## Data Availability

Data are contained within the article and available upon request.

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
