# Peer review of "Antiviral Activity of Anthranilamide Peptidomimetics against Herpes Simplex Virus 1 and a Coronavirus"

_antibiotics, 2023, doi:10.3390/antibiotics12091436_

Round 1
Reviewer 1 Report
1. In abstract, the initials of "Herpes Simplex Virus" are capital, which are different from the title. Please agree the full text.
2. Line 15 and 16, “-single stranded RNA virus ” and ”-double stranded DNA virus ” should be removed.
3. Line 20, the comma in "13 μ M, against HSV-1" should be removed. There are such problems elsewhere.
4. Line 35-37, the lack of ref. There are such problems elsewhere.
5. Line 65, the expression of "Herpes simplex infection" is incorrect.
6. Line 75, the disease name is not changed to the virus name.
7. In introduction, the content about coronavirus are more relevant, please reduce. HSV related content is a little less, please expand a little bit.
8. Line 144-145, the numbers in the "references 38 [39] and 39 [40]" are confusing.
9. In Figure 1 and 2, we can not understand the meaning of line and column wells in plaques test, and the ordinate in missing unit in figure 1, 2 and 3.
10. “pfu/mL” “PFU/mL” should be same.
11. Line 371, the beginning of the sentence should be retracted into three letters, or consistent with other places.
12. MHV-1 does not represent all RNA viruses, and HSV-1 does not represent all DNA viruses. Please change the relevant statements in the article.
13. The content of "2.1 Chemistry" can not be taken as a result. If taken as a result, it is contradictory to the content of the abstract and title, and can be elaborated in the preface or discussion section.
14. In Table 1, compounds 11 and 14 showed inhibitory activity against both MHV-1 and HSV-1, but results of TEM only was compound 14 for HSV-1 and compound 11 for MHV-1. Please increase the electron TEM results of compound 14 for MHV-1 and the TEM results of compound 11 for HSV-1. Although the TEM results for only two strains are not fully representative of all viruses, it is somewhat more convincing than a single one.
15. Compound 15,16 and 17 showed inhibitory activity against MHV-1 and compound 7 and 9 against HSV-1, but there are no corresponding results in the exploration of the inhibition mechanism and TEM assays. Please add the corresponding test. Then, the differences of compound structure on antiviral mechanisms could be compared.
1. In abstract, the initials of "Herpes Simplex Virus" are capital, which are different from the title. Please agree the full text.
2. Line 15 and 16, “-single stranded RNA virus ” and ”-double stranded DNA virus ” should be removed.
3. Line 20, the comma in "13 μ M, against HSV-1" should be removed. There are such problems elsewhere.
4. Line 35-37, the lack of ref. There are such problems elsewhere.
5. Line 65, the expression of "Herpes simplex infection" is incorrect.
6. Line 75, the disease name is not changed to the virus name.
7. In introduction, the content about coronavirus are more relevant, please reduce. HSV related content is a little less, please expand a little bit.
8. Line 144-145, the numbers in the "references 38 [39] and 39 [40]" are confusing.
9. In Figure 1 and 2, we can not understand the meaning of line and column wells in plaques test, and the ordinate in missing unit in figure 1, 2 and 3.
10. “pfu/mL” “PFU/mL” should be same.
11. Line 371, the beginning of the sentence should be retracted into three letters, or consistent with other places.
12. MHV-1 does not represent all RNA viruses, and HSV-1 does not represent all DNA viruses. Please change the relevant statements in the article.
13. The content of "2.1 Chemistry" can not be taken as a result. If taken as a result, it is contradictory to the content of the abstract and title, and can be elaborated in the preface or discussion section.
14. In Table 1, compounds 11 and 14 showed inhibitory activity against both MHV-1 and HSV-1, but results of TEM only was compound 14 for HSV-1 and compound 11 for MHV-1. Please increase the electron TEM results of compound 14 for MHV-1 and the TEM results of compound 11 for HSV-1. Although the TEM results for only two strains are not fully representative of all viruses, it is somewhat more convincing than a single one.
15. Compound 15,16 and 17 showed inhibitory activity against MHV-1 and compound 7 and 9 against HSV-1, but there are no corresponding results in the exploration of the inhibition mechanism and TEM assays. Please add the corresponding test. Then, the differences of compound structure on antiviral mechanisms could be compared.
Reviewer 2 Report
The article adequately describes the antiviral activity of peptidomimetic anthranilamide compounds against MHV-1 and HSV-1. However, some minor changes are required.
1. A brief introduction of chemistry of anthranilamide should be included in the introduction section.
2.Why have authors selected only anthranilamide for this study? This should be mentioned.
3. What was the reason of selection of MHV-1 and HSV-1 for antiviral study? This should be mentioned.
4. It is better to provide higher resolution picture of SEM.
------------------
Reviewer 3 Report
The authors investigated antiviral activities of 17 anthranilamide peptidomimetics against herpes simplex virus 1 and murine hepatitis virus 1. They concluded that Compounds 11 and 14 displayed the most potent inhibitory effects with IC50 values of 2.38 μM and 6.3 μM against MHV-1; while compounds 9 and 14 19 showed IC50 values of 14.8 μM and 13 μM against HSV-1. In general, the manuscript was well-written. It may be published pending some minor revisions.
Detailed comments:
1) The authors did not study SARS-CoV-2. So the word "coronavirus" may be misleading and should be replaced by "murine hepatitis virus 1"
2) Line 14 in Abstract "short anthranilamide-based peptidomimetic compounds 1-17" should be "seventeen short anthranilamide-based peptidomimetic compounds".
3) In Scheme 1 and Scheme 2, all abbreviations should be introduced with full names, such as "TFA" and "RT".
4) Some letters in Figure 1 are too small to be seen clearly.
5) Multiple comparisons should be applied for Figures 3. Or the significance should be marked.
Please pay attention to the plural of words, e.g. Line 19 "effects".
